# Flat-top TIRF illumination boosts DNA-PAINT imaging and quantification

Florian Stehr [1], Johannes Stein [1], Florian Schueder [1,2], Petra Schwille [1] & Ralf Jungmann [1,2]

Super-resolution (SR) techniques have extended the optical resolution down to a few nanometers. However, quantitative treatment of SR data remains challenging due to its complex dependence on a manifold of experimental parameters. Among the different SR variants, DNA-PAINT is relatively straightforward to implement, since it achieves the necessary 'blinking' without the use of rather complex optical or chemical activation schemes. However, it still suffers from image and quantification artifacts caused by inhomogeneous optical excitation. Here we demonstrate that several experimental challenges can be alleviated by introducing a segment-wise analysis approach and ultimately overcome by implementing a flat-top illumination profile for TIRF microscopy using a commercially-available beam-shaping device. The improvements with regards to homogeneous spatial resolution and precise kinetic information over the whole field-of-view were quantitatively assayed using DNA origami and cell samples. Our findings open the door to high-throughput DNA-PAINT studies with thus far unprecedented accuracy for quantitative data interpretation.

[1] Max Planck Institute of Biochemistry, 82152 Martinsried, Munich, Germany. [2] Faculty of Physics and Center for Nanoscience, Ludwig Maximilian University, 80539 Munich, Germany. These authors contributed equally: Florian Stehr, Johannes Stein. Correspondence and requests for materials should be addressed to P.S. (email: schwille@biochem.mpg.de) or to R.J. (email: jungmann@biochem.mpg.de)

The advent of super-resolution microscopy has revolutionized life science research by providing access to molecular structures with light microscopy, which were previously hidden below the diffraction limit. One of the major branches in the field is referred to as single molecule localization microscopy (SMLM) and includes methods such as photo-activated localization microscopy[1] (PALM), Stochastic optical reconstruction microscopy[2] (STORM), point accumulation in nanoscale topology[3] (PAINT), and their descendants[4]. In STORM and PALM, the blinking required for super-resolution reconstruction is obtained by complex photo-physical switching and activation of target-bound fluorophores. In contrast, PAINT imaging is based on reversible binding of a fluorescent species to the target structure. DNA-PAINT[5] exploits the specificity of DNA by using single-stranded oligonucleotides as labels ("docking strands") to which fluorescently-labeled complementary "imager" strands bind. Due to the non-fluorogenic nature of imagers (i.e., dye-labeled imager strands do fluoresce if not bound to their respective target strands), DNA-PAINT experiments are typically performed using some sort of selective plane illumination and/or detection, such as total internal reflection fluorescence (TIRF) microscopy[6], oblique illumination[7], or spinning disk confocal microscopy[8]. Besides offering spectrally-unlimited multiplexing capabilities (Exchange-PAINT)[9] and quantitative imaging (qPAINT)[10], DNA-PAINT can achieve spatial resolutions down to ~5 nm using standard TIRF microscopy[5]. As it is the case for all SMLM methods, reconstructed images have to be carefully interpreted, as they can be prone to artifacts arising e.g., from inhomogeneous illumination caused by the Gaussian laser profile[11,12]. This becomes especially important if localization datasets are used to extract quantitative information such as blinking kinetics, absolute molecule numbers, and other parameters beyond "just" binning of localizations to render qualitative images. Furthermore, inhomogeneous illumination can lead to spot-detection and fitting artifacts, ultimately resulting in a non-truthful reconstruction of the image data. One prominent example are false localizations originating from multiple active single emitters in a diffraction-limited area. A manifold of rather sophisticated methods and algorithms have been developed to deal with these multi-emitter localizations in SMLM data[13–17]. However, they are often not straightforward to implement or computationally intense. Approaches for obtaining homogenous illumination throughout the field-of-view should make it possible to use rather simple global thresholding algorithms to efficiently filter out these mislocalizations and omit them from downstream analysis.

While different solutions for uniform laser excitation have been proposed and applied to SMLM[18–20], these approaches negatively affect TIRF microscopy, due to their inherent reduction of spatial coherence[18,19]. Although coherent transformation of a Gaussian laser beam into a flat-top intensity profile by means of refractive beam-shaping has been pioneered decades ago[21,22], only very recently flat-top TIR illumination has been reported with the help of refractive beam-shaping elements, promising clear advantages regarding the interpretation of single molecule experiments[23] and their potential application to SMLM[24].

In this study, we identify imaging and quantification artifacts introduced by inhomogeneous sample illumination in DNA-PAINT. To achieve this, we present a novel processing metric based on analyzing radial image segments that allows us to quantitatively assess these artifacts and—at least to some extend —overcome the limitation of inhomogeneous sample illumination without the need for sophisticated post-processing of the data. In order to improve on that and to reduce the amount of post-processing necessary to achieve truthful representation of the data, we employ flat-top TIR illumination for DNA-PAINT

microscopy and demonstrate an increased homogeneity of almost all experimental observables when compared to standard Gaussian illumination. This has several implications: first, we achieved the same spot detection efficiency throughout the whole FOV (important for truthful SMLM reconstruction), thus eliminating the necessity for advanced spot finding algorithms, which take non-uniform illumination into account. Second, the uniformity of the excitation field allowed us to obtain accurate and precise binding time distributions for DNA-PAINT, independent of the position in the FOV. We used this predictability to demonstrate improved kinetic analysis of binding durations over the whole FOV. Third, we achieved uniform localization precision allowing spatial resolution better than 10 nm. Lastly, we find that homogeneous TIR excitation enables us to robustly identify multi-emitter localizations simply according to the number of photons detected. By exploiting the advantage of DNA-PAINT that sufficient sampling of the target structure is provided due to reversible binding of new imagers, we can afford to exclude all of these multi-emitter localizations detected by straightforward thresholding and thereby largely improve image quality for artifact-free quantitative statements without sophisticated image post-processing. Combining all advantages, we performed cellular DNA-PAINT imaging of the microtubule network in fixed cells and achieved a significant reduction of artifacts in the periphery compared to Gaussian illumination while preserving the image quality in terms of spatial resolution.

## Results

**Robust spot detection and homogeneous blinking.** To achieve flat-top illumination, we employed a refractive beam-shaping element called piShaper (AdlOptica GmbH, Berlin, Germany), which we placed in the excitation path of a custom-built TIRF microscope (a setup sketch can be found in Supplementary Figure 1). While transforming the profile of the excitation laser, refractive beam-shaping does preserve spatial coherence[23], which still enables efficient TIRF microscopy in contrast to previously reported flat-field super-resolution studies[18,19]. In order to quantitatively analyze the flat-top TIRF profile, we recorded a sequence of fluorescence images of a sample containing a high surface density of immobilized DNA origami structures, to which freely diffusing imager strands could bind. Figure 1a shows full-chip TIRF images obtained by averaging all acquisition frames for the Gaussian and flat-top profiles (left and right panel, respectively). Exemplary line profiles (Fig. 1b) show the fluorescence intensity variation along the specified axis for Gaussian (upper panel) and flat-top illumination (lower panel), yielding an intensity decrease by nearly a factor of three for Gaussian illumination vs. stable intensity for flat-top illumination.

In DNA-PAINT, blinking is achieved by the transient binding of short fluorescently-labeled DNA oligonucleotide "imager strands" to a DNA "docking" strand which is attached to the target of interest (Fig. 2a). The duration of blinking events is defined as bright time. We designed rectangular DNA origami nanostructures with a 20-nm-spaced pattern of $3 \times 4$ docking strands ("20-nm-grids", Fig. 2b) in order to quantitatively characterize the effect of inhomogeneous illumination on DNA-PAINT imaging. Super-resolution images of 20-nm-grids were acquired either using Gaussian or flat-top illumination and subsequently segmented into concentric rings such that each segment contained a similar number of structures (~800 per segment) for subsequent analysis (Fig. 2c).

First, we examined the detection efficiency of our spot finding and single-molecule fitting algorithm during SR reconstruction for a given threshold in the computed net gradient between adjacent pixels in the raw images[5]. Figure 2d compares exemplary

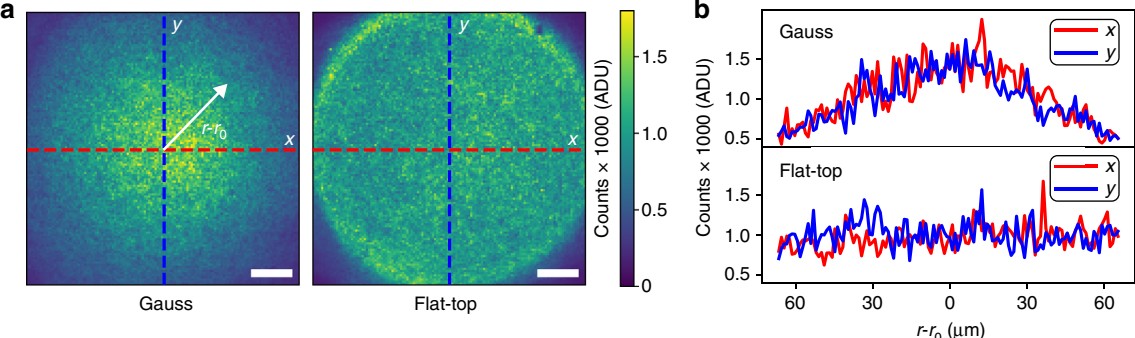

**Fig. 1** Gauss vs. flat-top illumination profiles. **a** Traditional illumination profile for TIRF microscopy with a Gaussian laser beam (left) compared to a flat-top profile created by a refractive beam-shaping device in the excitation path (right). **b** Line plots of fluorescence intensity along $x$ and $y$ axes (red and blue, respectively) for both profiles in **a**. Scale bars, 20 µm in **a**

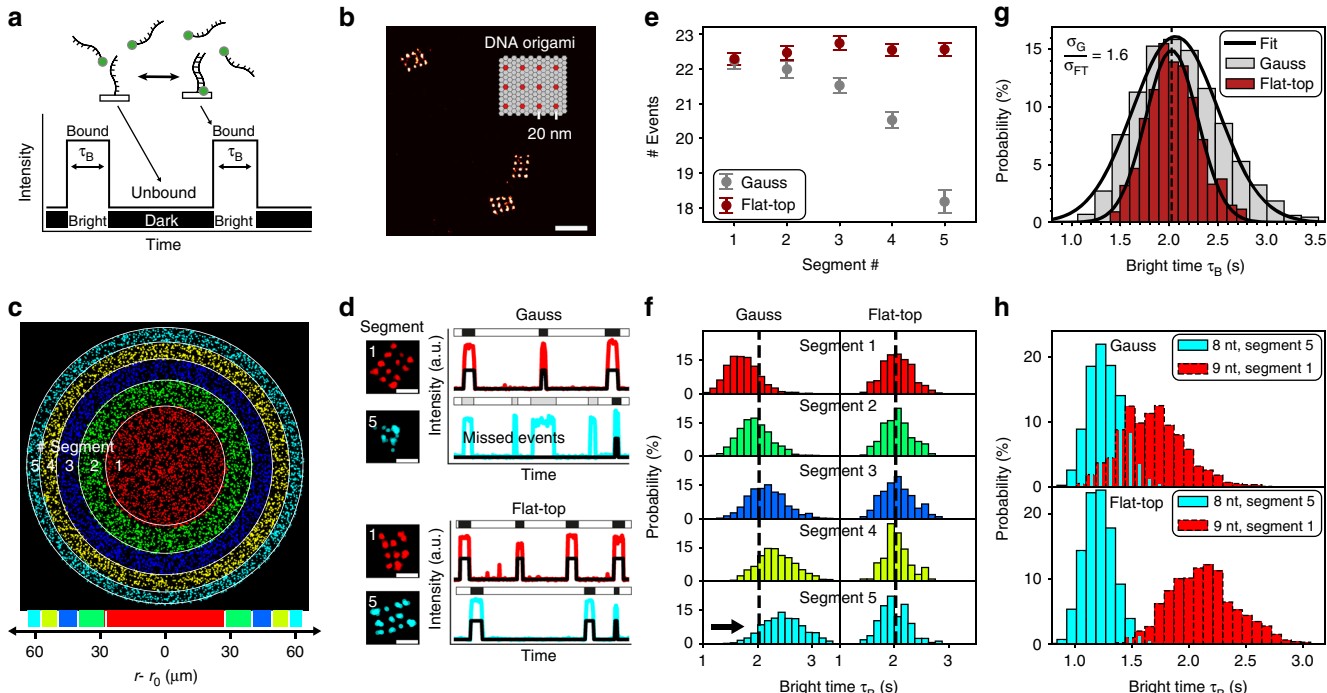

**Fig. 2** Flat-top illumination improves single-molecule detection and enables precise binding time quantification. **a** Schematic of DNA-PAINT: dye-labeled imager strands reversibly bind to complementary docking sites that are attached to the target of interest. Binding events result in apparent target blinking required for single molecule localization microscopy (SMLM). **b** DNA-PAINT image of rectangular DNA origami designed to display a 20-nm-grid pattern of docking strands (inset displaying origami design). **c** Whole-sCMOS-chip field of view (FOV) of several thousand DNA origami. Images acquired with Gaussian and flat-top illumination are both segmented into concentric rings containing equal numbers of origami (~800 origami per segment) for downstream quantification. **d** Exemplary DNA origami and intensity traces from inner and outer segment (red and cyan, respectively) showing that binding events in the outer segment are missed by the spot detection algorithm for Gaussian illumination. **e** The effect illustrated in **d** leads to a decrease in the mean number of binding events per origami with radial distance for the Gaussian profile. Flat-top illumination allows robust spot detection over the whole FOV. **f** Inhomogeneous photobleaching of imager strands increases the mean bright time with radial distance for Gaussian illumination. **g** The effect observed in **f** leads to an overall broadening of the bright time distribution over the whole FOV. **h** Distinction of docking strands of different length via bright times. Position-dependent bright times for Gaussian illumination lead to non-separable populations. Scale bars, 20 nm in **b** and 40 nm in **d**. Error bars in **e** correspond to SEM

intensity traces from 20-nm-grids in segments 1 and 5, highlighting that for Gaussian illumination blinking events in the outer segments were not detected anymore, resulting in poor sampling of the DNA origami image. This is due to the fact that the inhomogeneous profile of Gaussian illumination leads to a systematic decrease of the net gradient in DNA-PAINT raw images with increasing radial distance from the center (Supplementary Figure 2). The same effect was visible when comparing the average number of apparent binding events per 20-nm-grid between the segments (Fig. 2e). However, images acquired with flat-top illumination showed a constant net gradient resulting in a homogeneous spot detection efficiency (Fig. 2d and

Supplementary Figure 2) and ultimately in a constant number of binding events (Fig. 2e).

Next, we investigated the illumination effects on the bright times of imager binding events using our 20-nm-grids. As the localization precision in SMLM increases with the number of detected photons per acquisition frame[25,26], it is generally advisable to adapt camera integration times, dye switching duty cycles, and photon emission rates to obtain optimal localization precision. While the finite photon budget of fixed dyes in approaches like STORM or PALM sets a practical limit to the number of photons per switching cycle[27], PAINT-based approaches have the advantage that every blinking event originates from a "fresh" dye, thus the full photon budget of this dye can be harvested for superior localization precision. However, this comes at the cost of potentially bleaching a certain fraction of imager strands before they have dissociated from their targets. In order to enable precise adjustment of binding and bleaching times for e.g., qPAINT quantification, this bleaching probability should be constant over the FOV. For a Gaussian illumination profile, we observed that imager strands (9 nucleotides in length) binding to the center of the field of view photobleach faster than in the outer segments, as one would expect (Fig. 2f). In contrast, images acquired with flat-top illumination exhibited homogeneous bright times for the same imager species throughout the FOV. The radial bright time dependence for Gaussian illumination resulted in a broadening of the total bright time distribution over the FOV by a factor of $\sigma_G/\sigma_{FT} = 1.6$ compared to flat-top illumination (Fig. 2g). Inhomogeneous bleaching conditions have direct implications for quantitative statements based on the blinking kinetics from DNA-PAINT images. Figure 2h shows that for DNA-PAINT images of 20-nm-grids with either shorter-binding 8 nucleotide-long (nt) or longer-binding 9-nt-long docking strands acquired with the same imager under identical conditions, it was not possible to distinguish between the two bright time populations comparing segments 1 and 5 for Gaussian illumination (but it still allows for differentiation within each segment, see Supplementary Figure 3a). However, flat-top illumination allowed us to clearly separate bright time distributions over the full FOV. Analogously to Fig. 2g the total bright time distributions for both 8-nt and 9-nt 20-nm-grids are narrower for flat-top illumination (Supplementary Figure 3b). Enhanced control over the bleaching behavior allowed us to both resolve single 20-nm-grid structures (see Supplementary Figure 4) and simultaneously distinguish between short and long binding duration with high fidelity.

**Uniform localization precision and mislocalization filtering**. In order to obtain a measure of how precise a single DNA-PAINT docking strand could be localized, we used a previously developed "averaging" tool in Picasso that allowed us to pick all 20-nm-grids in an image and to align them onto a model grid[28]. Figure 3a displays the averaged images of more than 700 structures each from segments 1 (red) and 5 (cyan) for the same sample imaged with Gaussian and flat-top illumination (a $20 \times 20$ subset of individual 20-nm-grid images can be found in Supplementary Figure 5). The histograms represent the spatial distribution of localizations along the dashed lines. A double Gaussian fit recovered the designed docking strand spacing of ~20 nm. The evident loss of resolution in the Gaussian average from segment 5 compared to segment 1 is confirmed by the broadened peaks in the histograms which increased by almost a factor of two (localization precision from ~2.0 to ~3.5 nm). On the contrary, in the flat-top image only a minor decrease in localization precision was observed (~10%). As previously mentioned, the localization uncertainty in SMLM is inversely proportional to the square-root

of the number of detected photons. We identified a three-fold decrease in the average number of detected photons per localization event from ~15,000 to ~5,000 comparing segments 1 and 5 for Gaussian illumination and attributed this as the main cause for the decrease in localization precision (Fig. 3b). Segment-wise calculation of the localization precision based on Nearest Neighbor Analysis[5,29] (NeNA) confirmed this relation (Fig. 3c). Nevertheless, we also observed a radial decrease in photon number and localization precision for the image acquired with flat-top illumination. Since this effect is decoupled from the excitation profile, we attribute this to finite aperture effects that become increasingly apparent in the periphery when increasing the FOV. However, this only leads to minor radial performance and image resolution loss (~10 %) compared to the performance decrease due to inhomogeneous excitation in the case of Gaussian illumination.

In order to benchmark the overall localization precision for flat-top illumination, we designed and imaged DNA origami structures with a 10-nm-grid pattern of docking strands. We could resolve the individual docking strands even in segment 5, demonstrating better than 10 nm spatial resolution over the entire FOV, ~130 μm in diameter (Fig. 3d).

Straightforward filtering capabilities during image post-processing are an additional advantage of using flat-top illumination. Figure 3e depicts the photon count distribution for a 20-nm-grid sample imaged with Gaussian (top) and flat-top illumination (bottom). In contrast to Gaussian illumination, we were able to identify two distinct peaks in the distribution from the image acquired with the flat-top profile. The first peak at 25,800 photons is attributed to localizations originating from single imager binding events. The second peak is located at roughly twice the number of photons (53,200) and represents localizations originating from two imager strands bound simultaneously to the same structure. The top panel in Fig. 3f illustrates that these multi-emitter events result in mislocalizations, thus degrading the overall image quality. In contrast to the Gaussian profile (only in segment 1 the photon count distribution indicates two peaks, see Supplementary Figure 6), flat-top illumination allowed us to robustly use an upper threshold limit over the whole FOV at the $1/e^2$ value of the first peak for filtering out these mislocalizations during post-processing and thereby considerably improving the quality of the super-resolved image (Fig. 3f, bottom).

**Improved large field-of-view cellular imaging with DNA-PAINT**. After identifying and quantifying the effects caused by inhomogeneous illumination on DNA origami structures, we applied flat-top illumination for imaging cellular structures with DNA-PAINT to highlight the differences in obtainable overall image quality researchers should expect on common samples. Figure 4a shows SR images of the microtubule network in fixed COS-7 cells labeled using primary and DNA-conjugated secondary antibodies[5,30] and subsequent DNA-PAINT imaging for Gaussian (left) and flat-top illumination (right) acquired with the full camera sensor resulting in a field-of-view of $130 \times 130 \, \mu m^2$. The magnified regions in the center and the border of the image (segment 1 and 5 as defined in Fig. 2c) recorded using Gaussian illumination show an increasing loss of localizations towards the periphery due to the limited spot-detection efficiency (see Fig. 4b, bottom left). In contrast, we obtain a uniform localization density using flat-top illumination, confirming the earlier observations for DNA origami experiments (Fig. 4b, right. Find a detailed two-level zoomed cell image in Supplementary Figure 7). The white arrows point to regions of accumulated multi-emitter mislocalizations in between the densely-labeled microtubules (for

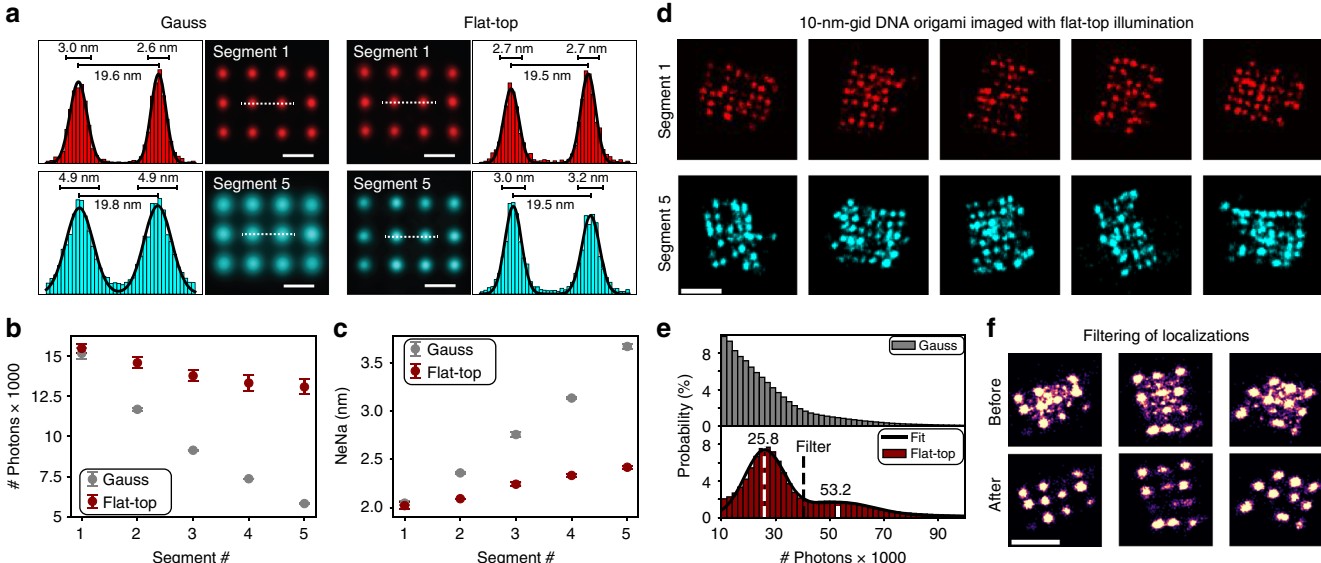

**Fig. 3** Localization precision of 2 nm over 130 × 130 μm² FOV with flat-top illumination. **a** Averaged images of 20-nm-grid structures (~800 per segment, see Fig. 2c for definition of segments) show radial decrease in resolution using Gaussian illumination, while flat-top illumination maintains high spatial resolution. Fit results for peak-to-peak distance and standard deviation displayed above. **b** Mean number of detected photons per localization per frame. **c** Localization precision calculated by nearest neighbor analysis (NeNA) **d** 10-nm-grid DNA origami design for whole-chip resolution benchmarking under flat-top illumination. **e** Photon count histogram for flat-top illumination indicating two peaks for the case of single binding and simultaneous binding events of two imager strands to a 20-nm grid. **f** Filtering out simultaneous binding events above single binding threshold (filter in **e** set 1/e² value of first distribution) allowing the removal of "cross talk" localizations in between two active docking strands. Scale bars, 20 nm in **a**, 50 nm in **d** and **f**. Error bars in **b** and **c** correspond to SEM

magnified illustration see Supplementary Figure 8). These could again be identified in all photon count histograms in Fig. 4c, except for segment 5 of the image acquired using Gaussian illumination. Figure 4d demonstrates the gain in image quality for both segments of the flat-top image after removal of all localizations above the 1/e² value of the single emitter peak. The distributions of localizations in the boxed regions along the indicated directions in Fig. 4d show two distinct peaks originating from the 2D projection of a homogenously-labeled rod. Even in the periphery of the full camera sensor image we recovered a peak-to-peak distance of ~37 nm which is in good agreement with previously reported values from SR studies[5,19,30,31]. Despite the radial quality loss in the image acquired with Gaussian illumination, we could also identify and remove multi-emitter mislocalizations in the center of the image (Supplementary Figure 9).

Overall, high-throughput DNA-PAINT SMLM employing large FOVs can hence benefit from flat-top illumination without substantial trade-off in image quality.

## Discussion
We here presented a quantitative super resolution study of flat-top TIRF illumination for DNA-PAINT. We demonstrated that flat-top illumination improves the quantification accuracy in DNA-PAINT data by enabling both homogeneous spatial resolution and precise kinetic blinking parameters over large FOVs. In addition, uniform illumination gives rise to new features in the experimental observables, that can be used during straightforward post-processing. This includes a more robust spot detection and enabled us to effectively remove multi-emitter artifacts without the use of computationally demanding multi-emitter localization algorithms[13–17]. We achieved the latter by simple photon number thresholding in the resulting localization datasets. We want to note, that using this threshold to omit multi-emitter mislocalizations does not necessarily lead to a reduced image quality due to missed localizations in DNA-PAINT, as we can collect a considerably

larger amount of total localizations per docking strand due to the repetitive nature of image acquisition.

Furthermore, improved control over the photobleaching conditions allowed us to distinguish apparent identical structures of different docking strand length independent of their position within the FOV. This could be exploited for non-spectral multiplexing in DNA-PAINT super resolution microscopy in the future. We think that these numerous advantages will significantly enhance the statistical treatment of single-molecule microscopy data, since a flat-top illumination allows the use of the complete FOV for further analysis and can hence pave new routes for high-throughput experiments. Furthermore, a uniform TIR excitation will improve single-molecule fluorescence-based binding affinity studies on surfaces, e.g., by SI-FCS[32], since photophysical effects can be treated globally and can therefore be decoupled from local changes caused by other physical effects. In cases where phototoxicity has to be minimized[33], flat-top illumination can provide precise control over the whole FOV.

Regarding the comparison of Gaussian and flat-top illumination several of our findings can also directly improve image quality for quantitative DNA-PAINT with a Gaussian excitation profile, when segment-wise analysis of parameters is employed. With regard to biological samples, however, segmented analysis will presumably be most beneficial in the case of compact, separable protein structures such as nuclear pore complexes compared to continuous networks such as the cytoskeleton or large organelle structures. Using this segmentation approach, we showed that in the center segment it is also possible to remove multi-emitter localizations for more precise and quantitative data interpretation. Furthermore, the differentiation between structures with short and long binding docking strands is also possible within each segment, but obviously this comes at the cost that the overall statistics is divided by the number of introduced segments.

In conclusion however, we are convinced that the advantages arising from flat-top TIR illumination—especially with regards to

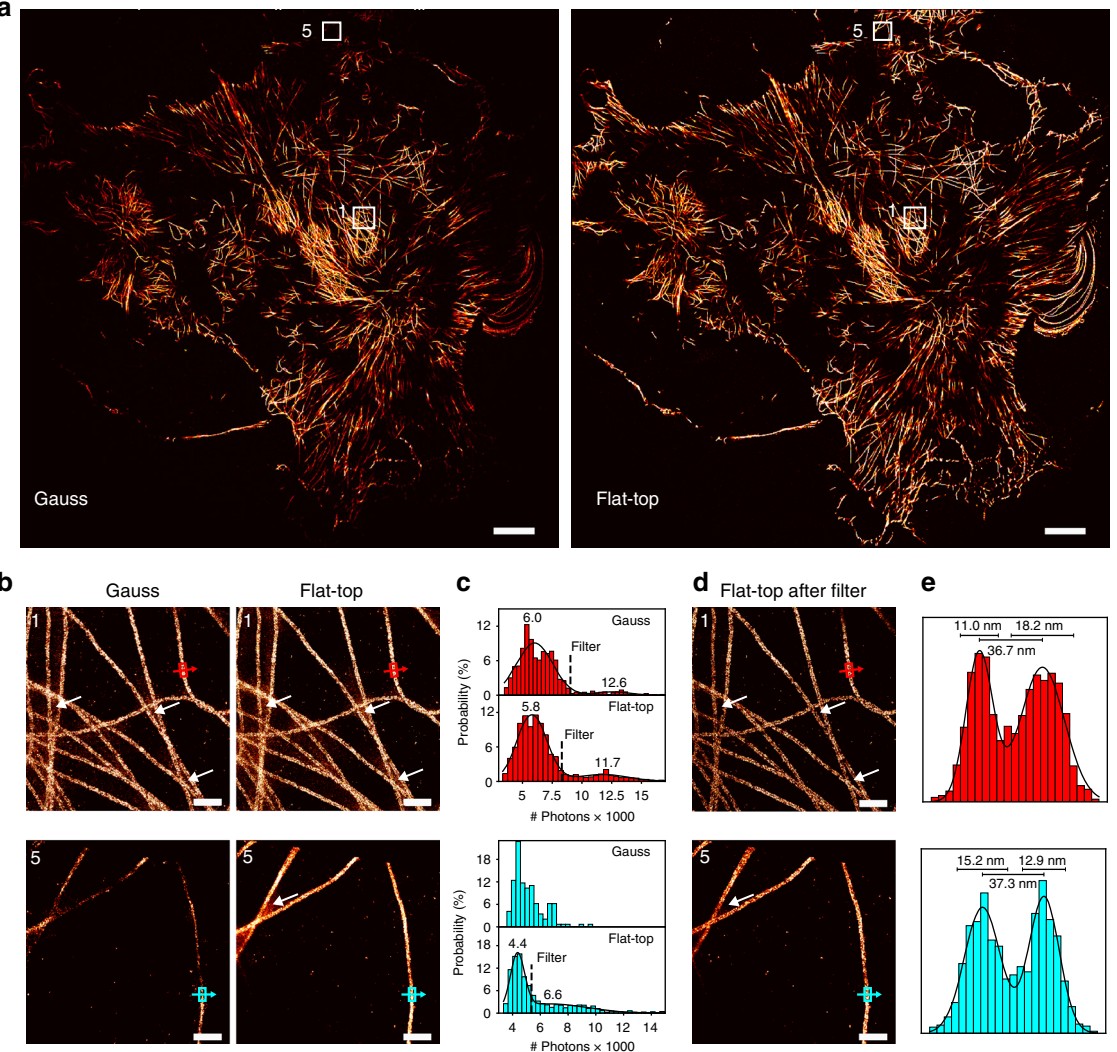

**Fig. 4** Artifact removal for uniform and quantitative cellular DNA-PAINT imaging. **a** Full camera chip (130 × 130 μm²) DNA-PAINT image of the microtubule network in fixed COS-7 cells acquired using Gauss illumination (left) and the same field of view for flat-top illumination (right). **b** Magnified sections from segment 1 and segment 5 (as defined in Fig. 2a) highlighting the image quality in the center and the border region of the camera chip. White arrows point to artifacts due to multi-emitter mislocalizations. **c** Photon count histograms for box regions in images from **b**. Double Gaussian fit allows identification and removal of multi-emitter mislocalizations (threshold at 1/e² of first peak) except for segment 5 for Gaussian illumination. **d** Filtered flat-top images from **b** displaying enhanced image quality after removing mislocalization artifacts. **e** Intensity profiles across single microtubules indicated in **d**. Scale bars, 10 μm in **a**, 500 nm in **b**, **d**

the ease-of-use and availability of commercial beam shaping devices—are clearly superior and we believe they might become a standard feature for TIRF microscopy.

## Methods

**Materials.** Unmodified, dye-labeled, and biotinylated DNA oligonucleotides were purchased from MWG Eurofins. DNA scaffold strands were purchased from Tilibit (p7249, identical to M13mp18). Streptavidin (cat: S-888) and glass slides (cat: 10756991) were ordered from Thermo Fisher. Coverslips were purchased from Marienfeld (cat: 0107052). PEG-8000 was purchased from Merck (cat: 6510-1KG). Tris 1M pH 8.0 (cat: AM9856), EDTA 0.5M pH 8.0 (cat: AM9261), Magnesium 1M (cat: AM9530G) and Sodium Chloride 5M (cat: AM9759) were ordered from Ambion. Ultrapure water was obtained from a Milli-Q filter machine. Tween-20 (cat: P9416-50ML), Glycerol (cat: G5516-500ML), Methanol (cat: 32213-2.5L), BSA-Biotin (cat: A8549), Protocatechuate 3,4-Dioxygenase Pseudomonas (PCD) (cat: P8279), 3,4-Dihydroxybenzoic acid (PCA) (cat: 37580-25G-F) and (+-)-6-Hydroxy-2,5,7,8-tetra-methylchromane-2-carboxylic acid (Trolox) (cat: 238813-5G) were ordered from Sigma-Aldrich. Twinsil two-component glue was purchased from Picodent (cat: 13001000). Monoclonal antibodies against alpha-tubulin (cat: MA1-80017) was purchased from Thermo Scientific. The secondary

antibodies Anti-Rat (cat: 712-005-150) were purchased from Jackson ImmunoResearch.

**Buffers.** Five buffers were used for sample preparation and imaging: Buffer A (10 mM Tris-HCl pH 7.5, 100 mM NaCl, 0.05% Tween 20, pH 7.5); Buffer B (5 mM Tris-HCl pH0 8, 10 mM MgCl2, 1 mM EDTA, 0.05% Tween 20, pH 8); Buffer C (1× PBS pH 8, 500 mM NaCl, pH 8); 100× Trolox: 100 mg Trolox, 430 μl 100% methanol, 345 μl of 1M NaOH in 3.2 ml H2O. 40× PCA: 154 mg PCA, 10 ml water, and NaOH were mixed and adjusted to pH 9.0. 100× PCD: 9.3 mg PCD, 13.3 ml of buffer (100 mM Tris-HCl pH 8, 50 mM KCl, 1 mM EDTA, 50% glycerol).

**DNA origami design, assembly, and purification.** DNA origami structures were designed using the design module of Picasso[5] (see Supplementary Figure 10). Folding of structures was performed using the following components: single-stranded DNA scaffold (0.01 μM), core staples (0.5 μM), biotin staples (0.5 μM), modified staples (each 0.5 μM), 1× folding buffer in a total of 50 μl for each sample. Annealing was done by cooling the mixture from 80 to 25 °C in 3 h in a thermocycler. Structures were purified using PEG-precipitation[34].

**DNA origami sample preparation**. A glass slide was glued onto a coverslip with the help of double-sided tape (Scotch, cat. no. 665D) to form a flow chamber with inner volume of ~20 μl. First, 20 μl of biotin-labeled bovine albumin (1 mg/ml, dissolved in buffer A) was flushed into the chamber and incubated for 2 min. The chamber was then washed with 40 μl of buffer A. Twenty microliter of streptavidin (0.5 mg/ml, dissolved in buffer A) was then flushed through the chamber and incubated for 2 min. After washing with 40 μl of buffer A and subsequently with 40 μl of buffer B, 20 μl of biotin-labeled DNA structures (1:80 dilution in buffer B from purified DNA-origami stock) were flushed into the chamber and incubated for 10 min. The chamber was washed with 40 μl of buffer B. Finally, 40 μl of the imager solution was flushed into the chamber, which was subsequently sealed with two-component glue before imaging. A list of all staples can be found in Supplementary Tables 1 and 2.

**Cell sample preparation**. COS7 cells were cultured with Eagle's minimum essential medium fortified with 10% FBS with penicillin and streptomycin and were incubated at 37 °C with 5% $CO_2$. At ~30% confluence, cells were seeded into Eppendorf 8-well chambered cover glass ~24 h before fixation and were grown to ~70% confluence. For fixation, the samples were pre-fixed and pre-permeabilized with 0.4% glutaraldehyde and 0.25% Triton X-100 for 90 s. Next, the cells were quickly rinsed with 1× PBS once followed by fixation with 3% glutaraldehyde for 15 min. Afterwards, samples were rinsed twice (5 min) with 1× PBS and then quenched with 0.1% $NaBH_4$ for 7 min. After rinsing four times with 1× PBS for 30 s, 60 s, and twice for 5 min, samples were blocked and permeabilized with 3% BSA and 0.25% Triton X-100 for 2 h. Then, samples were incubated with 10 μg/ml of primary antibodies (1:100 dilution) in a solution with 3% BSA and 0.1% Triton X-100 at 4 °C overnight. Cells were rinsed three times (5 min each) with 1× PBS. Next, they were incubated with 10 μg/ml of labeled secondary antibodies (1:100 dilution) in a solution with 3% BSA and 0.1% Triton X-100 at room temperature for 1 h. For fiducial based drift correction, the samples were incubated with gold nanoparticles with a 1:1 dilution in 1× PBS for 5 min. Finally, samples were rinsed three times with 1× PBS before adding imager solution.

**Super-resolution microscopy setup**. Fluorescence imaging was carried out on an inverted custom-built microscope (see setup sketch in Supplementary Figure 1) in an objective-type TIRF configuration with an oil-immersion objective (Olympus UAPON, 100×, NA 1.49). One laser was used for excitation: 561 nm (1 W, DPSS-system, MPB). Laser power was adjusted by polarization rotation with a half-wave plate (Thorlabs, WPH05M-561) before passing a polarizing beam-splitter cube (Thorlabs, PBS101). To spatially clean the beam-profile the laser light was coupled into a single-mode polarization-maintaining fiber (Thorlabs, P3-488PM-FC-2) using an aspheric lens (Thorlabs, C610TME-A). The coupling polarization into the fiber was adjusted using a zero-order half wave plate (Thorlabs, WPH05M-561). The laser light was re-collimated after the fiber using an achromatic doublet lens (Thorlabs, AC254-050-A-ML) resulting in a collimated FWHM beam diameter of ~6 mm. The laser light was split into two paths of approximately equal length using a combination of two flip mirrors (Thorlabs, FM90/M). In one path the laser light was unaltered resulting in a Gaussian beam profile for excitation. In the other path a diffractive beam shaper device (AdlOptica, piShaper 6_6_VIS) transformed the Gaussian beam profile in a collimated flat-top profile. Both paths were realigned to each other and passed the same downstream optics. Switching between the two illumination schemes can therefore be achieved by flipping two mirrors simultaneously. The laser beam diameter was magnified by a factor of 2.5 using a custom-built Telescope (Thorlabs, AC254-030-A-ML and Thorlabs, AC508-075-A-ML). The laser light was coupled into the microscope objective using an achromatic doublet lens (Thorlabs, AC508-180-A-ML) and a dichroic beam splitter (AHF, F68-785). Fluorescence light was spectrally filtered with an emission filter (AHF Analysentechnik, 605/64) and imaged on an sCMOS camera (Andor, Zyla 4.2) without further magnification (Thorlabs, TTL180-A) resulting in an effective pixel size of 130 nm (after 2 × 2 binning). Microscopy samples were mounted on a x-y-z stage (ASI, S31121010FT and ASI, FTP2050) that was used for focusing with the microscope objective being at fixed position. Our custom TIRF setup was used for all Figures.

**Imaging conditions**. All fluorescence microscopy data was recorded on the full sensor (2048 × 2048 pixels, pixel size: 6.5 μm) of our SCMOS camera operated with the open source acquisition software μManager[35] at a read out rate of 200 MHz and a dynamic range of 16 bit. Detailed imaging conditions for all main and supplementary figures can be found in Supplementary Table 3. The laser power refers to the power measured after the fiber (see Supplementary Figure 1). As can be seen in Fig. 1b, the mean intensity of the flat-top profile is at around 60 % of the Gaussian peak intensity, when operated at the same power. Supplementary Figure 11 illustrates that by an respective power increase we can adjust the flat-top profile to the Gaussian peak intensity. Sequence design of imager and docking strands can be found in Supplementary Table 4.

**Super-resolution reconstruction**. Raw fluorescence data was subjected to spot-finding and subsequent super-resolution reconstruction using the localize module of the Picasso software package[5]. Localizations were then loaded into Picasso's

render module and drift-corrected. DNA origamis were automatically selected using the "Pick similar" function with the following settings: pick radius: 143 nm; standard deviation: 1.5, 1.7, 1.9 (subsequently). After automated selection, picked areas were saved as "Picked localizations" for further processing.

**Kinetic analysis**. Picasso's render module[5] allows automatic recognition of ROIs within the rendered super-resolution image by searching for similarity in the localization distribution to pre-selected user defined regions of specific size. The resulting ROIs of the complete set of localizations are referred to as "picks" (Supplementary Figure 12a). We calculated characteristic quantities associated with the temporal distribution of localization events within each of these picks with a custom written python script (see Supplementary Figure 12). Since the automated selection of ROIs cannot distinguish between repetitive (specific) and non-repetitive (unspecific) blinking behavior we implemented a filtering procedure based on the temporal distribution of localization events.

**Filtering**. By looking at the temporal distribution of the localization events (trace) associated to a single pick we can define its mean and standard deviation. We refer to these parameters as the mean (localization) frame and its standard deviation (std) in the units of frames. Repetitive transient binding to DNA origami throughout the measurement leads hence to a mean (localization) frame of roughly half the number of total frames in the acquisition window with a large standard deviation (Supplementary Figure 12b, left panel). In contrast non-repetitive binding will result in a mean (localization) frame located within the frames of their unique occurrence randomly distributed throughout the acquisition window and a small standard deviation (Supplementary Figure 12b, right panel). Plotting the distribution of the mean (localization) frame and its standard deviation over all automatically selected ROIs thus allows clear identification of a major population of picked areas showing repetitive blinking while outliers indicating non-repetitive blinking behavior can be disregarded for further analysis (Supplementary Figure 12c and d).

**Averaging**. Picked origami structures were averaged to a designed model structure using the average3 module of Picasso with a pixel oversampling of 40 and setting a custom symmetry of 180 degree[28].

**Reporting summary**. Further information on experimental design is available in the Nature Research Reporting Summary linked to this article.

**Code availability**. All code supporting the findings of this study is available from the corresponding author upon request.

## Data availability
The data that support the findings of this study are available from the corresponding authors upon reasonable request.

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

## Acknowledgements
We thank Julian Bauer, Patrick Schueler, Bianca Sperl and Sigrid Bauer for experimental support. F.St., and J.S. acknowledge support by the QBM graduate school and the Center for Nanoscience Munich. This work has been supported in part by the German Research Foundation through the Emmy Noether Program (DFG JU 2957/1-1 to R.J.), the SFB1032 (projects A11 and A09 to R.J. and P.S.), the European Research Council through an ERC Starting Grant (MolMap; grant agreement number 680241 to R.J.) and the Max Planck Society (P.S. and R.J.).

## Author contributions
F.St. and J.S. built the microscope, designed and performed experiments, analyzed data, and wrote the manuscript. F.Sc. designed DNA origami structures, performed cell experiments and wrote the manuscript. F.St., J.S., F.Sc., and R.J. conceived of the study. R. J. supervised the study, interpreted data, and wrote the manuscript. P.S. supervised the study and wrote the manuscript. All authors reviewed and approved the manuscript.

## Additional information

**Competing interests:** The authors declare no competing interests.

