## [Peer Review File · Nature Communications]

Reviewers' comments:

Reviewer #1 (Remarks to the Author):

I have carefully evaluated the manuscript entitled "Flat-top TIRF illumination boosts DNA-PAINT imaging and quantification" by Stehr et al. Overall, the manuscript is well written, the analyses are well executed and the manuscript is probably worthy of publication. I am, however, concerned about the degree of novelty that is reported in this manuscript and I don't think that in its current form it is worthy of publication in Nature Communications, but rather in a journal with a lower impact factor, e.g. Scientific Reports. My reasoning for this is as follows:

1. The simple addition of a beam shaping device to homogenize the excitation in TIRF-based single molecule localization microscopy is not significant enough per se and the effects of the homogenized excitation of fluorescence as detailed by the authors are largely expected. As the authors have pointed out themselves, a few groups have already used (albeit) different methods for homogenizing the excitation field for flat-fielded excitation in single molecule fluorescence microscopy. The simple addition of another, commercially available, flat-fielding device is, in my opinion, not significant enough, plus other, previous attempts of flat-field, e.g. by Axelrod (probably the "father" of TIRF microscopy) by beam-scanning, as well as the well-known ring TIRF illumination scheme by the group of Derek Toomre were ignored. Furthermore, the reasoning for the need for flat-fielding is flawed. The authors appear to imply that coherent illumination is somehow required for TIRF illumination (e.g. on page 3), when in fact it is not. Coherent excitation might improve the efficiency of TIRF excitation, but the TIRF condition can be achieved with any form of light, even fully incoherent light, and, indeed, many commercial adapters for TIRF illumination in fluorescence microscopy don't necessarily require laser excitation.

2. In the second sentence on page 2, it is argued that due to the "non-fluorogenic" nature of imagers, some form of selective plane illumination is required?? Obviously, this should say "fluorogenic nature"? The main reason why selective plane, and in particular TIRF illumination, is required is that there are many imager strands freely diffusing in the background, which need to somehow be excluded from the excitation.

3. In my opinion, the only major new result of this manuscript is the observation that flat-fielding improves the photon statistics to the point where it can be used to effectively and efficiently excluded multiple localizations (see Fig. 3f) without the need for sophisticated multi-peak fitting algorithms. If one considers how much time and effort is still spent in trying to utilize data from multiple localizations of densely labeled samples by developing even more sophisticated methods, see e.g. the recent "HAWK" procedure by the group of Susan Cox in Nature Methods, then anything that helps improve this situation by simpler means is worth reporting. So, if the authors could refocus their paper on this application and elaborate on it, then I could still be convinced that the paper could be worthy of publication in Nature Communications, but, in its current form I would have to recommend that the paper be rejected and published in another, lower impact journal.

Reviewer #2 (Remarks to the Author):

In their manuscript "Flat-top TIRF illumination boosts DNA-PAINT imaging and quantification" Stehr et al present the application of a recently first presented refractive approach to provide uniform illumination for TIRF to single molecule localisation microscopy, specifically DNA-PAINT.

The MS makes a number of interesting points.

1) In comparing the current widely used TIRF approaches that have an essentially Gaussian

profile, the authors introduce an interesting segmented analysis for the Gaussian case.

It appears to this reviewer that several of the advantages that the MS puts forward as a result of the uniform illumination intensity, can be achieved by applying the proposed criteria in segments, as already done in some of the comparisons, i.e. (1) effective removal of double binding events via a simple photon number criterion, (2) binding time based distinction of docking strands. The criteria would have to be chosen by segment but this is clearly not computationally expensive nor complex. In this regard it seems to me that some of the advantages appear overstated and it should rather positively be stated that a segment based analysis can provide a fairly straightforward improvement of Gaussian illumination as well.

Clearly, the uniform illumination makes this more convenient and a larger field of view is available with high signal-to-noise ratio data.

2) A couple of consequences of the insertion of the beam-shaping device should be more clearly elaborated on:

- how much does the insertion of the device reduce the peak intensities than can be reached? this could be based on comparison to the segmented analysis, by stating which segment has equivalent illumination intensity for same laser power and the intensity ratios for the other segments.

- does the slightly lower peak intensity that can be reached still allow maximal harvesting of photons from DNA-PAINT imagers in the scenarios tested?

- how does the device affect the HILO mode which is also of high interest, particularly for biological samples. The supplementary has images for this configuration but the main text lacks any comment on the uniformity/non-uniformity of the resulting distribution.

3) The binding time based analysis has a number of features that should be clarified (material relating to Supp Fig. 10 and corresponding text areas):

- I found the terminology not particularly well chosen and hard to follow until I had unpacked what terms such as 'mean frame time' and 'pick' meant. Both seem to me non-standard and not very helpful without clearer definition.

- assuming a pick refers to a small, and often sparse ROI in the data, possibly selected in software via a feature of the resulting rendered super-resolution image. Then the analysis appears to relate to the events associated with this ROI or 'pick' and their distribution in time is analysed. The 'mean frame time' is then the mean of the time distribution of events, where the natural time unit of SMLM data is the frame number in which an event was detected. The width of this time distribution, measured by its standard deviation, is then the second measure used. I would find it useful if a clear plain word definition of the terms used (without reference to specific software etc) can be provided, as I have attempted here.

- it would be helpful to actually show a couple of time distributions, say in sup fig 10, that show examples of the cases mentioned (mean in centre, mean elsewhere, small width)

- can such analysis be used for non-sparse regions in biological samples, e.g. within the areas covered by microtubules; it seems that the analysis suggested has a sparsity requirement that is well fulfilled with e.g. origami, but probably not with many biological samples

- I briefly looked at the software linked from the text to do the pick-based analysis (<https://github.com/DerGoldeneReiter/qPAINT>). It was not clear which scripts were used in the

set of files and directories in the distribution. Could an example with a test data set be provided for clarification?

Reviewer #1 (Remarks to the Author):

I have carefully evaluated the manuscript entitled "Flat-top TIRF illumination boosts DNA-PAINT imaging and quantification" by Stehr et al. Overall, the manuscript is well written, the analyses are well executed and the manuscript is probably worthy of publication. I am, however, concerned about the degree of novelty that is reported in this manuscript and I don't think that in its current form it is worthy of publication in Nature Communications, but rather in a journal with a lower impact factor, e.g. Scientific Reports. My reasoning for this is as follows:

We thank the reviewer for the constructive criticism of our work and we are happy to address all points in our response to his specific comments below. We also refocused the paper on the quantitative aspects of our analysis approach and included new data on cellular structures in Figure 4 highlighting the straightforward implementation of our simple thresholding-based removal of multi-emitter mislocalizations.

1. The simple addition of a beam shaping device to homogenize the excitation in TIRF-based single molecule localization microscopy is not significant enough per se and the effects of the homogenized excitation of fluorescence as detailed by the authors are largely expected. As the authors have pointed out themselves, a few groups have already used (albeit) different methods for homogenizing the excitation field for flat-fielded excitation in single molecule fluorescence microscopy. The simple addition of another, commercially available, flat-fielding device is, in my opinion, not significant enough, plus other, previous attempts of flat-field, e.g. by Axelrod (probably the "father" of TIRF microscopy) by beam-scanning, as well as the well-known ringTIRF illumination scheme by the group of Derek Toomre were ignored. Furthermore, the reasoning for the need for flat-fielding is flawed. The authors appear to imply that coherent illumination is somehow required for TIRF illumination (e.g. on page 3), when in fact it is not. Coherent excitation might improve the efficiency of TIRF excitation, but the TIRF condition can be achieved with any form of light, even fully incoherent light, and, indeed, many commercial adapters for TIRF illumination in fluorescence microscopy don't necessarily require laser excitation.

*The reviewer is correct with regards to his statement that a coherent light source is not a necessary requirement for TIRF illumination per se, however it improves the efficiency. We now have rephrased our statement in the main text to reflect this. The sentence now reads: "While different solutions for uniform laser excitation have been proposed and applied to SMLM, these approaches **negatively affect** TIRF microscopy, due to **their inherent reduction of spatial coherence**". We stand by our argument that for highest quality TIRF illumination (which is essential for TIRF-based SMLM), spatial coherence is a requirement in order to focus the beam tightly to the back-focal plane of the objective (see also Khaw, I. et al. Flat-field illumination for quantitative fluorescence imaging. *Opt. Express* 26, 15276–15288 (2018)).*

We thank the reviewer for pointing us to more previous attempts to achieve more homogenous illumination in TIRF microscopy by Axelrod and Toomre. However, as we understand it, the use of beam-scanning or ringTIRF illumination per se does not lead to flat-top illumination, but allows to remove typical TIRF "interference fringes" by the spinning of the TIRF beam (see also doi: 10.1016/S0091-679X(08)00607-9). We would be happy to include more references and discuss the approaches the reviewer mentioned in light of our manuscript, if the reviewer would kindly point us to the specific publications dealing with flat-top illumination.

2. In the second sentence on page 2, it is argued that due to the "non-fluorogenic" nature of imagers, some form of selective plane illumination is required?? Obviously, this should say "fluorogenic nature"? The main reason why selective plane, and in particular TIRF illumination, is required is that there are many imager strands freely diffusing in the background, which need to somehow be excluded from the excitation.

We apologize for not making this point clearer and creating confusion about the term “non-fluorogenic”. We have now added a sentence explaining the meaning of “non-fluorogenic” in this context: Dye-labeled imager strands do also emit fluorescence when they are not bound to their respective target strands, as the reviewer correctly pointed out.

The term "fluorogenic" is typically used for probes that are not currently fluorescent but will become so, e.g. upon binding to a target molecule of interest. This is a term that is widely accepted in the field of fluorescent labels, see e.g. <https://doi.org/10.1038/nmeth.2972> and others.

3. In my opinion, the only major new result of this manuscript is the observation that flat-fielding improves the photon statistics to the point where it can be used to effectively and efficiently excluded multiple localizations (see Fig. 3f) without the need for sophisticated multi-peak fitting algorithms. If one considers how much time and effort is still spent in trying to utilize data from multiple localizations of densely labeled samples by developing even more sophisticated methods, see e.g. the recent "HAWK" procedure by the group of Susan Cox in Nature Methods, then anything that helps improve this situation by simpler means is worth reporting. So, if the authors could refocus their paper on this application and elaborate on it, then I could still be convinced that the paper could be worthy of publication in Nature Communications, but, in its current form I would have to recommend that the paper be rejected and published in another, lower impact journal.

Again, we do apologize to have created the impression that the major innovation in the paper is the use of a refractive beam-shaping device, when in fact it is not.

We thank the reviewer for his suggestion to refocus the paper more on the quantitative analysis capability we introduced for photon statistics. We used his advice and have now generally rewritten the manuscript to highlight more the quantitative nature of the data obtainable with flat-field illumination.

We specifically focus on the point that our segment-wise analysis can be used to compare areas of similar excitation intensity even in the case of Gaussian illumination. We furthermore discuss now in greater detail the advantages for simple post-processing after flat-field illumination without the need for sophisticated algorithms. We have replaced Figure 4 with cellular microtubule data highlighting the case of thresholded photon statistics analysis in order to remove multi-emitter artifacts in the dataset.

Reviewer #2 (Remarks to the Author):

In their manuscript “Flat-top TIRF illumination boosts DNA-PAINT imaging and quantification” Stehr et al present the application of a recently first presented refractive approach to provide uniform illumination for TIRF to single molecule localisation microscopy, specifically DNA-PAINT.

The MS makes a number of interesting points.

1) In comparing the current widely used TIRF approaches that have an essentially Gaussian profile, the authors introduce an interesting segmented analysis for the Gaussian case.

It appears to this reviewer that several of the advantages that the MS puts forward as a result of the uniform illumination intensity, can be achieved by applying the proposed criteria in segments, as already done in some of the comparisons, i.e. (1) effective removal of double binding events via a simple photon number criterion, (2) binding time based distinction of docking strands. The criteria would have to be chosen by segment but this is clearly not computationally expensive nor complex. In this regard it seems to me that some of the advantages appear overstated and it should rather positively be stated that a segment based analysis can provide a fairly straightforward improvement of Gaussian illumination as well.

We thank the reviewer for the supportive comment with regards to our segment-wise analysis approach. We have now refocused the paper to better highlight this aspect and included additional data analysis (see Supplementary Figs. 3, 6, and 9) for the case of Gaussian illumination. We show that indeed (1) effective removal of double binding events is possible via our simple photon number criterion in the center segment for the Gaussian illumination case, and (2) binding time-based distinction of docking strands within respective segments is possible for 8 vs. 9 nt long docking strands.

We apologize for potentially overstating some of the advantages, and now hope that the revised version represents a more truthful analysis of these points. We are also grateful to the reviewer for the suggestion to focus more on the segmented analysis in the Gaussian case and hope that the revised manuscript makes this now clearer, more accessible and beneficial for researchers relying on Gaussian illumination only.

However, we are still convinced (and our data supports this in our opinion), that flat-field illumination is clearly advantageous, as our segment-wise analysis cannot deal with all disadvantages of non-uniform excitation: E.g. the robust identification of double-binding events is only possible in the center segment in the Gaussian illumination case (see Supplementary Fig. 6, 9, and also Figure 4). Furthermore, the decrease in localization precision due to lower excitation power densities in the outer part of the image will lead to reduced spatial resolution and cannot be recovered by the segment analysis. This can only be achieved by the flat-top illumination. Lastly, the global analysis and comparison of binding times will become impossible if larger statistics is necessary or the sample has varying spatial densities of objects of interest. This is also not a problem for the flat-field illumination.

Clearly, the uniform illumination makes this more convenient and a larger field of view is available with high signal-to-noise ratio data.

2) A couple of consequences of the insertion of the beam-shaping device should be more clearly elaborated on:

- how much does the insertion of the device reduce the peak intensities than can be reached? this could be based on comparison to the segmented analysis, by stating which segment has equivalent illumination intensity for same laser power and the intensity ratios for the other segments.

We now include a more detailed analysis about the reduction in peak intensities when comparing the center of the Gaussian illumination case with the flat-top case given the same input power. This is detailed in Supplementary Figure 11. In brief, the mean intensity of the flat-top profile is at around 60 % of the Gaussian peak intensity, when operated at the same input power.

- does the slightly lower peak intensity that can be reached still allow maximal harvesting of photons from DNA-PAINT imagers in the scenarios tested?

Supplementary Fig. 11 also illustrates that by a respective power increase we can adjust the flat-top profile to the Gaussian peak intensity and thus make full use of the photon budget for DNA-PAINT microscopy.

- how does the device affect the HILO mode which is also of high interest, particularly for biological samples. The supplementary has images for this configuration but the main text lacks any comment on the uniformity/non-uniformity of the resulting distribution.

We have now removed the HILO data in the supplement. This is a result of the request from reviewer 1, that the manuscript should be refocused towards the improvement in dealing with multi-emitter localizations and more quantitative treatment of the data. We initially presented the supplementary figure to support our

claim that TIRF can still be efficiently achieved with the beam-shaping device. However, due to the refocus of the manuscript, we have removed this figure. That being said, we thank the reviewer for this note and think that HILO is indeed not affected by the beam-shaping device.

3) The binding time based analysis has a number of features that should be clarified (material relating to Supp Fig. 10 and corresponding text areas):

- I found the terminology not particularly well chosen and hard to follow until I had unpacked what terms such as 'mean frame time' and 'pick' meant. Both seem to me non-standard and not very helpful without clearer definition.

- assuming a pick refers to a small, and often sparse ROI in the data, possibly selected in software via a feature of the resulting rendered super-resolution image. Then the analysis appears to relate to the events associated with this ROI or 'pick' and their distribution in time is analysed. The 'mean frame time' is then the mean of the time distribution of events, where the natural time unit of SMLM data is the frame number in which an event was detected. The width of this time distribution, measured by its standard deviation, is then the second measure used. I would find it useful if a clear plain word definition of the terms used (without reference to specific software etc) can be provided, as I have attempted here.

- it would be helpful to actually show a couple of time distributions, say in sup fig 10, that show examples of the cases mentioned (mean in centre, mean elsewhere, small width)

We apologize for the missing clarification of the terminology used in the binding time analysis. We have now rephrased the sections in the Materials and Methods according to the suggestions of the reviewer. We have also added Supplementary Figure 11 to further illustrate the meaning of the respective parameters.

- can such analysis be used for non-sparse regions in biological samples, e.g. within the areas covered by microtubules; it seems that the analysis suggested has a sparsity requirement that is well fulfilled with e.g. origami, but probably not with many biological samples

We thank the reviewer for pointing us to this question. Indeed, such analysis might be tricky to implement in many biological samples. These types of segmented analysis would work well e.g. in the case of compact, separable protein structures such as nuclear pore complexes, however will most likely fail to produce satisfactory results in the case of e.g. cytoskeleton or organelle structures.

- I briefly looked at the software linked from the text to do the pick-based analysis (<https://github.com/DerGoldeneReiter/qPAINT>). It was not clear which scripts were used in the set of files and directories in the distribution. Could an example with a test data set be provided for clarification?

We apologize for the unclear documentation of the software used for data analysis. We now include a .zip file containing a test data file and the required custom python modules for our kinetic analysis (please find the zip file here: <https://www.dropbox.com/s/9of4ivm3z8avef4/code.zip?dl=0>). The code is now thoroughly commented, and we also added a readme file for instructions regarding usage.

REVIEWERS' COMMENTS:

Reviewer #1 (Remarks to the Author):

The authors have significantly modified their original document and satisfied most of my original concerns. I have no objections to the publication of this manuscript, but would like to suggest a few more additional (minor) modifications:

1) As the authors pointed out in their response to my original comments, the approaches taken by Axelrod and Toomre and colleagues do not provide flat-fielding. Instead they are used to "even out" and "homogenize" TIRF illumination. To credit Toomre for this development, I would suggest adding the following citation to the references:

Yang Q, Karpikov A, Toomre D, Duncan JS. 3-D reconstruction of microtubules from multi-angle total internal reflection fluorescence microscopy using Bayesian framework. *IEEE Trans Image Process.* 2011; 20(8):2248–2259

2) I think it would be wise to prominently discuss the fact that the improved segmented analysis does not apply to the majority of biological samples, in order to avoid that some researchers who might have just casually read the manuscript begin to pursue this in their research. A statement similar to the one provided in response to reviewer 2: "These types of segmented analysis would work well e.g. in the case of compact, separable protein structures such as nuclear pore complexes, however will most likely fail to produce satisfactory results in the case of e.g. cytoskeleton or organelle structures." would be beneficial - either in the introduction or in the conclusions to the paper.

Reviewer #1 (Remarks to the Author):

The authors have significantly modified their original document and satisfied most of my original concerns. I have no objections to the publication of this manuscript, but would like to suggest a few more additional (minor) modifications:

We thank the reviewer for the positive evaluation of the revised version of our manuscript.

1) As the authors pointed out in their response to my original comments, the approaches taken by Axelrod and Toomre and colleagues do not provide flat-fielding. Instead they are used to "even out" and "homogenize" TIRF illumination. To credit Toomre for this development, I would suggest adding the following citation to the references:

Yang Q, Karpikov A, Toomre D, Duncan JS. 3-D reconstruction of microtubules from multi-angle total internal reflection fluorescence microscopy using Bayesian framework. IEEE Trans Image Process. 2011;20(8):2248–2259

Thank you for this comment. We have now added the reference.

2) I think it would be wise to prominently discuss the fact that the improved segmented analysis does not apply to the majority of biological samples, in order to avoid that some researchers who might have just casually read the manuscript begin to pursue this in their research. A statement similar to the one provided in response to reviewer 2: "These types of segmented analysis would work well e.g. in the case of compact, separable protein structures such as nuclear pore complexes, however will most likely fail to produce satisfactory results in the case of e.g. cytoskeleton or organelle structures." would be beneficial - either in the introduction or in the conclusions to the paper.

We thank the reviewer for this suggestion and have now added a discussion to clarify this point in the main text.